# Diagnosis of Eosinophilic Otitis Media Using Blood Eosinophil Levels

**DOI:** 10.3390/diagnostics13233598

**Published:** 2023-12-04

**Authors:** Yeonsu Jeong, Gina Na, Jong-Gyun Ha, Dachan Kim, Junyup Kim, Seonghoon Bae

**Affiliations:** 1Department of Otorhinolaryngology, Severance Hospiral, Yonsei University College of Medicine, Seoul 03722, Republic of Korea; 2Department of Otorhinolaryngology—Head and Neck Surgery, Gwangmyeong Hospital, Chung-Ang University College of Medicine, Gwangmyeong 14353, Republic of Korea; 3Department of Physical Medicine and Rehabilitation, Hanyang University Medical Center, Seoul 04763, Republic of Korea; 4Department of Otorhinolaryngology, Gangnam Severance Hospital, Yonsei University College of Medicine, Seoul 06273, Republic of Korea

**Keywords:** eosinophils, eosinophilic otitis media, chronic rhinosinusitis, granulomatosis with polyangiitis, primary ciliary dyskinesia

## Abstract

Eosinophilic otitis media (EOM) is a rare middle ear disease with unfavorable outcomes. Under the current diagnostic criteria of EOM, it is challenging to suspect EOM before tympanostomy. Therefore, this study attempted to use blood eosinophil levels for the differential diagnosis of EOM from other conditions. Three disease groups with features of recurrent otorrhea were categorized, which included the following: EOM (*n* = 9), granulomatosis with polyangiitis (GPA, *n* = 12), and primary ciliary dyskinesia (PCD, *n* = 6). Clinical and radiological characteristics were analyzed in the three groups. Patients who underwent ventilation tube insertion due to serous otitis media were enrolled as the control group (*n* = 225) to evaluate the diagnostic validity of blood eosinophilia. The EOM group showed a significantly higher blood eosinophil concentration (*p* < 0.001) and blood eosinophil count (*p* < 0.001) compared to the GPA and PCD groups. The estimated sensitivity and specificity for diagnosing EOM from OME patients who underwent ventilation tube insertion were 100% and 95.6%, respectively. In addition, EOM tended to have protympanic space soft tissue density and a relatively clear retrotympanic space in temporal bone computerized tomography. Blood eosinophil evaluation is a significant clinical indicator of EOM. Furthermore, the assessment of exclusive protympanic soft tissue density can provide an additional diagnostic clue.

## 1. Introduction

Eosinophilic otitis media (EOM) is a rare middle ear disease with unfavorable outcomes. It does not respond to conventional treatment, including surgery. Moreover, approximately 50% of patients with EOM have sensorineural hearing loss [1]. EOM was first described by H. Nagamine et al. in 2001, and Y. Iino et al. further established its diagnostic criteria in 2011 [1,2]. EOM has significantly more frequent associations with various typical clinical characteristics than ordinary otitis media. In EOM patients, there is an association with bronchial asthma in 90%, resistance to conventional treatment in 93%, viscous middle ear effusion in 93%, and nasal polyposis in 62% of patients [1]. These are the minor criteria of EOM suggested by Y. Iino et al. Most importantly, as a major criterion, patients with EOM have eosinophil dominant middle ear effusion. In addition, in EOM patients, bilaterality (83%) and chronic rhinosinusitis (CRS, 74%) were also significantly higher than those in the control group [1].

EOM must be differentially diagnosed from several diseases that have similar symptoms with a poor treatment response. Anti-neutrophil cytoplasmic autoantibody (ANCA)-related vasculitis syndrome, such as Wegener’s granulomatosis (granulomatosis with polyangiitis, GPA) and the Churg–Strauss syndrome (eosinophilic granulomatosis with polyangiitis, EGPA), has also been reported to present with intractable otitis media [3]. In addition, otitis media with ANCA-associated vasculitis, also called OMAAV, was recently introduced for categorizing recurrent otitis media with AAV without fulfilling systemic vasculitis [4,5]. Primary ciliary dyskinesia (PCD) is also known to present with recurrent otitis media with effusion [6]. These disease entities also tend to have concurrent chronic sinusitis and other problems involving the respiratory system. Under the current diagnostic criteria of EOM, there are no options to exclude EOM before tympanostomy to confirm the viscosity and eosinophil count of the effusion.

Therefore, we tried to apply the blood eosinophil level to patients with EOM, GPA, PCD, and otitis media with effusion (OME) requiring ventilation tube insertion to narrow the diagnosis of EOM. In addition, the radiological characteristics of patients with EOM were also investigated with computerized tomography (CT). With this study, we inform clinicians of the necessity of middle ear sampling before encountering it in the surgical field. Moreover, this study presents the association of systemic eosinophilia in EOM, which has previously rarely been discussed in depth.

## 2. Materials and Methods

### 2.1. Patient Enrollment

This study was conducted using a retrospective medical chart review. In total, four groups (EOM, PCD, GPA, and control group) of patients were enrolled in this study. The patients who (1) were diagnosed with EOM, PCD, and GPA, and (2) visited an ENT specialist to diagnose otitis media were selected after searching the lead author’s hospital database from 1 November 2005 to 1 September 2021. The inclusion criteria of EOM were consistent with the diagnostic criteria of EOM, which were suggested by Y. Iino (especially including dominant eosinophils in middle ear effusion). In PCD, patients with pathological confirmation via nasal mucosa biopsy were included. In GPA, only ANCA-positive patients were included. The exclusion criteria were as follows: (1) absence of a complete blood count (CBC) test within ±2 weeks from the diagnosis of otitis media, (2) absence of imaging studies that visualize maxillary and ethmoid sinuses, and (3) absence of nasal endoscopic findings in the medical chart. None of the 27 patients met the exclusion criteria.

The control group consisted of OME patients who underwent ventilation tube insertion and who were found in the same database and the same period. The search conditions were as follows: (1) ventilation tube insertion operation, (2) the presence of a CBC study result 2 weeks before ventilation tube insertion, (3) age ≥20 years, and (4) the operative note describing the presence of serous middle ear effusion. A total of 517 patients were found and included in the control group. There were no overlapping patients in the EOM, PCD, GPA, and the control group. The protocol of this study was approved (project number: 4-2022-0209) by the institutional review board of Severance Hospital (Seoul, Republic of Korea), and the need for informed consent was waived due to the retrospective nature of this study.

### 2.2. Evaluation of Eosinophilia and CT Scan

The eosinophil concentration was adopted from the nearest CBC study to the diagnosis of otitis media, which was conducted within ±2 weeks. The first head and neck imaging study (including sinus CT and temporal bone CT) in the database was analyzed to evaluate nasal conditions, such as polyps and sinusitis. The radiographic paranasal sinus and Water’s views were analyzed in two patients with PCD who had not undergone a CT scan.

The first temporal bone CT in the database was analyzed to evaluate the middle ear condition of each disease group. The involvement was determined using the consensus of two otology specialists (G.N. and S.B.) in a dichotomous fashion, and it was labeled “clear” or “involved”. The pneumatization of the mastoid was graded according to the system suggest by Han et al. [7]. The protympanum and retrotympanum were classified based on the malleus handle on the temporal bone CT.

### 2.3. Acquisition and Preprocessing of Temporal Bone Computed Tomography Scan and Lesion Segmentation

Patients were scanned using a Siemens SOMATOM Force multidetector scanner (Definition Flash, Siemens Healthcare, Forcheim, Germany) with identical settings. Routine protocols of temporal bone CT scans were as follows: axial plane, matrix = 512 × 512, voxel size = 0.4043 × 0.4043 × 0.5000 mm^3^, 96 slices.

The regions of interest (ROIs) surrounding lesions were drawn by an otolaryngology specialist (S.B) on each slice using 3D-slicer (https://www.slicer.org/, accessed on 1 December 2022) software ver 4.11.20210226. Everyone involved in this process was blinded to the clinical data. The CT scans and ROIs were then reoriented and normalized to a standard MNI152 template using the Statistical Parametric Mapping (SPM) software, version 12 (http://www.fil.ion.ucl.ac.uk/spm/, accessed on 1 December 2022) [8]. The threshold of the spatially normalized CT scans was set to an attenuation level of 405, and the scans were superimposed and binarized for the generation of study-specific skull templates. Then, among the normalized ROIs, the ROI of the right middle ear was flipped to the left and overlaid with the ROI originally on the left to visualize the lesion distribution. All preprocessing steps except for lesion delineation were performed by a researcher specializing in neuroimaging analysis (J.K.).

### 2.4. Statistical Analysis

Pearson’s chi-square test (two-tailed) was used to evaluate the significance of different proportions of multiple groups. Analysis of variance (ANOVA) with the Holm–Sidak post hoc multiple comparisons test was used to evaluate the differences in continuous values between the multiple groups after the Shapiro–Wilk test for normality. If the values did not pass the normality test, the Kruskal–Wallis test with Dunn’s post hoc multiple comparisons test (or Mann–Whitney test) were used to analyze the continuous values for multiple groups. SPSS 25.0 (IBM, Armonk, NY, USA) and Prism 8.0 (GraphPad Software, San Diego, CA, USA) were used for statistical analyses. Average values are represented as means ± standard deviations (SD). Results with *p*-values less than 0.05 were considered statistically significant.

## 3. Results

### 3.1. Differential Diagnosis of EOM, GPA, and PCD

The onset age of middle ear disease was significantly earlier in the PCD group (*p* = 0.027 and 0.011 compared to EOM and GPA, respectively) (Table 1). By contrast, the EOM and GPA groups developed middle ear disease in middle age. The GPA group showed significantly fewer incidents of nasal polyposis (16.7%, *p* = 0.001) and bilateral otitis media (33.3%, *p* = 0.004) compared to the EOM and PCD groups. Most of the enrolled patients (26 of 27) were found to have CRS. We further investigated the blood eosinophils in each enrolled patient (Figure 1). The blood eosinophil concentration (*p* < 0.001 and *p* = 0.003 for GPA and PCD, respectively) and count (*p* < 0.001 and *p* = 0.002 for GPA and PCD, respectively) were significantly higher in the EOM group than in the other groups. The sensitivity and the specificity for diagnosing EOM were both 100% in all enrolled patients when the cut-off values of blood eosinophil concentration and blood eosinophil count were 6.0% and 450/μL, respectively. 

### 3.2. Diagnostic Performance of Blood Eosinophilia for EOM Compared to OME

To exclude possible EOM, we selected OME patients with serous effusion confirmed by the surgeon as the control group (Table 2). In the 225-patient OME group, 23 (10.2%) patients had an eosinophil concentration higher than 6.0%, and 13 (5.8%) had an eosinophil count higher than 450/μL. In addition, 10 (4.4%) patients had increased results in both parameters (eosinophil concentration > 6.0% and eosinophil count > 450/μL). Accordingly, when both eosinophil concentration (cut-off value as 6.0%) and eosinophil count (cut-off value as 450/μL) were used for diagnosis, the specificity was 95.6% and the sensitivity was 100%. 

### 3.3. Diagnostic Performance of Blood Eosinophilia for EOM Compared to OME

In addition, we analyzed the radiological characteristics of EOM, GPA, and PCD patients for additional clinical clues. Two children in the PCD group and three patients in the GPA group did not undergo a temporal bone CT, and one patient in the EOM group had a post-mastoidectomy status decades ago. After excluding these patients, temporal bone CT images of 41 ears were subsequently analyzed. The CT images of the EOM group showed a distinctive feature of soft tissue density in the protympanum (near the Eustachian tube orifice and anterior to the promontory) (Figure 2 and Figure A1). The retrotympanum and mastoid cavity were less likely to be affected compared to the protympanum. Although there was no statistical significance (*p* = 0.096) in the proportion involving the protympanum between the groups, we found a significantly higher likelihood (*p* = 0.008) of having a soft tissue density exclusively in the protympanum of the EOM group (Table 3). The exclusive involvement of the protympanum was also significantly more frequent (*p* = 0.005) when compared to the OME group (Table A1). Interestingly, the PCD group showed significantly worse conditions in the mastoid cavity. All ears with PCD showed a soft tissue density in the mastoid (*p* = 0.009) with a severely sclerotic mastoid cavity (*p* < 0.001). In addition, the PCD group also had the highest chance of retrotympanum involvement (*p* = 0.029).

## 4. Discussion

This study suggests that EOM can be preliminarily diagnosed based on blood eosinophilia before choosing to proceed with its pathological confirmation. A combined cut-off value of 6% in eosinophil concentration and a 450/μL eosinophil count showed a sensitivity of 100% and a specificity of 95.6% for diagnosing EOM among the control group consisting of OME patients who underwent ventilation tube insertion. In addition, the CT finding of soft tissue density exclusively in the protympanum was found in 41.2% of patients with EOM and was scarce in the patients with GPA (5.6%), PCD (0%), and OME (4.5%). Using these clinical cues, EOM can be distinguishable from the other diseases with refractory otitis media as well as OME, which is the most frequent.

Given the systemic symptoms of EOM, it is likely to present as a sub-symptom of a systemic disease. Aspirin-exacerbated respiratory disease (AERD), classically known as Samter’s triad, is reported to be related to nasal polyps in 9.7% and asthma in 7.2% of patients [9]. Ear symptoms also have a high prevalence in patients with AERD. Among these patients, 6.1% have chronic ear drainage, and 8.9% have a ventilation tube in their tympanic membrane [10]. Interestingly, AERD also has characteristic hyper-eosinophilia, both in the circulation and in tissues, and it typically develops in adulthood [11,12,13]. The overlapping symptoms and signs of AERD and EOM imply that these diseases are intimately related to each other. In this respect, our results’ indication that the blood eosinophilia predicts EOM seems plausible.

GPA and PCD showed several distinguishable characteristics compared to EOM not only in the eosinophil-related factors but also in the demographic and anatomical findings. The mastoid cavity of PCD was significantly sclerotic. The poor development of the mastoid cavity may relate to the early development of otitis media in PCD when compared to GPA and EOM. Since PCD is a congenital disease that is related to a higher incidence of pneumonia and sinusitis due to the poor mucociliary clearance, the mean age of diagnosis is below 10 years in Europe [14]. The higher chance of retrotympanic space and mastoid cavity involvement may arise from the poor clearance function of the Eustachian tube and middle ear epithelium. However, not all PCD is diagnosed at a young age [15]; in this study, 50% of PCD patients were diagnosed at age 30 or older. PCD should be considered as a possible diagnosis for recurrent otitis media that should be differentially diagnosed with EOM.

Temporal bone CT of EOM showed exclusive protympanum soft tissue density, which is the first to be reported, to the best of our knowledge. Previously, a case report described the radiological characteristics of EOM [16]. The clearly visible Eustachian tube was a major characteristic of EOM in that case report. In contrast, our results showed that 58.8% of patients with EOM had an obstructed Eustachian tube. The discrepancy may be due to the individual personal judgments made in these two studies. The different severity levels of the disease can also bias the radiological results. The radiological characteristics of EOM may be caused by the remarkably high viscosity of the middle ear effusion. The relatively clear (41.2%) Eustachian tube area (middle ear orifice) can be explained by the robust mucociliary function of the Eustachian tube epithelium, which consists of over 80% ciliated cells, and the funnel-shaped anatomy that can evoke the venturi effect according to Bernoulli’s equation [17,18]. Simply put, the viscous mucus is stuck in the area preceding the Eustachian tube, where it can resist mucociliary clearance and airflow into the nasal cavity. On the other hand, it may be related to the pathophysiological mechanism of EOM. Iino et al. suggested that an antigenic material invades the middle ear via the patulous Eustachian tube [2,19,20]. This pathogenesis can also explain the relatively clear retrotympanic space, since the protympanic space is anatomically closer to the orifice of the patulous Eustachian tube. In both cases, Eustachian tube balloon dilation should be avoided as obstructive Eustachian tube dysfunction seems to not be related to the pathogenesis of EOM. Theoretically, symptoms of obstructive Eustachian tube dysfunction could also be caused by soft tissue filling the middle ear orifice of the Eustachian tube. Therefore, the results of this study imply that the clinician should rule out EOM before deciding on Eustachian tube balloon dilation, which is considered for the treatment of recurrent otitis media with effusion.

This study had several limitations. First, the disease groups had a small number of patients due to the low incidence of EOM, GPA, and PCD. Nevertheless, given the results of this study, blood eosinophils should be recognized as a crucial marker for differential diagnosis of EOM from other diseases, since EOM has distinctive blood eosinophilia. A second limitation was that patients with EGPA, which also possibly presents as a high concentration of eosinophils in the middle ear discharge and peripheral blood, were not enrolled in this study. Furthermore, as the study design was a retrospective medical chart review, a scheduled imaging study and blood sampling were absent. In future studies, serial evaluation with treatment trials for EOM should be conducted. In an additional limitation, the sensitivity and specificity might have been biased, as the 225 OME patients in the control group who underwent ventilation tube insertion did not undergo histological confirmation of middle ear effusion to determine the presence of high levels of eosinophils. In addition, the small number of enrolled EOM patients might have produced an overestimate of the sensitivity. However, given that the high viscosity of the middle ear effusion has been accepted as a specific characteristic of EOM, the possibility that EOM patients were accidentally included in the control group consisting of OME with serous effusion is very low. Importantly, histopathological confirmation of middle ear effusion should still be required to diagnose EOM until a further large prospective study is conducted to analyze the correlation between current diagnostic criteria and the blood eosinophil count.

## 5. Conclusions

Patients with EOM showed a significantly higher peripheral blood eosinophil count and concentration compared to those with GPA, PCD, and OME. In addition, exclusive protympanic space involvement in CT scans may be another clue for diagnosing EOM. Therefore, blood eosinophil evaluation and CT scans are recommended to diagnose EOM in patients with refractory otitis media.

## Figures and Tables

**Figure 1 diagnostics-13-03598-f001:**
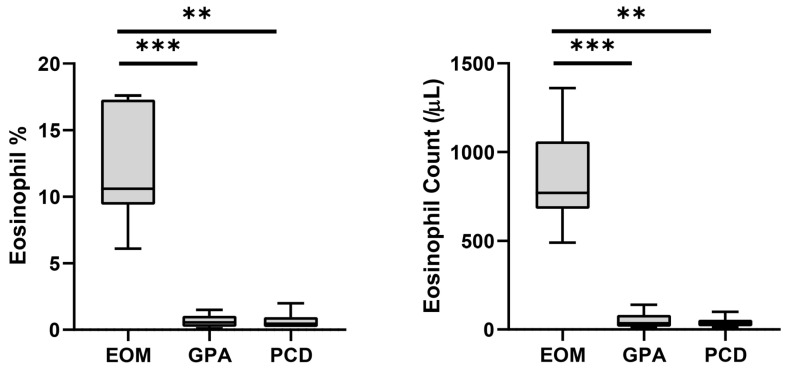
Peripheral blood eosinophil concentration and count of each disease group. Bars and error bars indicate median and range, respectively. Grey box indicates interquartile range. The Kruskal–Wallis test and Dunn’s multiple comparisons test were used for statistical analysis. (EOM: eosinophilic otitis media; GPA: granulomatosis with polyangiitis; PCD: primary ciliary dyskinesia; ***: *p* < 0.001; **: *p* < 0.01.)

**Figure 2 diagnostics-13-03598-f002:**
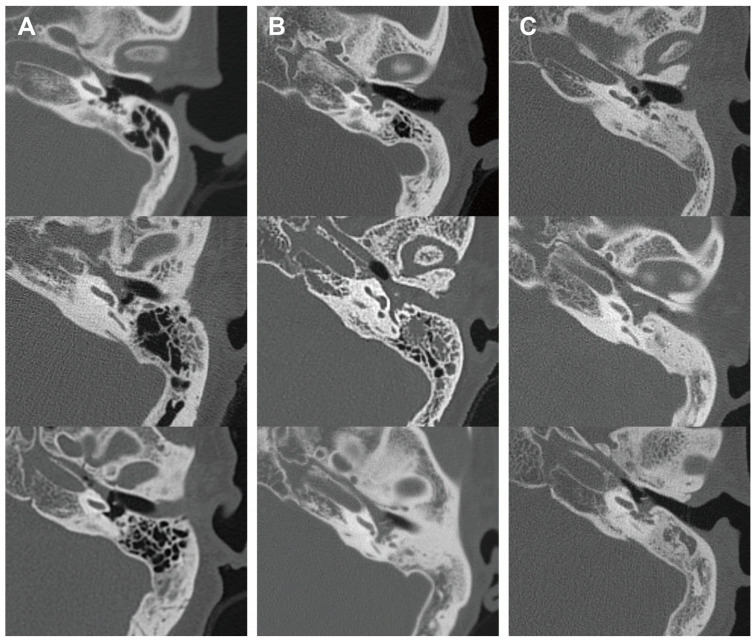
Representative middle ear images in temporal bone computerized tomography of each disease group. Axial images at malleus umbo level are presented. Columns (**A**–**C**) indicate EOM, GPA, and PCD groups, respectively. EOM group showed soft tissue density in protympanic space and relatively clear retrotympanic space. (EOM: eosinophilic otitis media; GPA: granulomatosis with polyangiitis; PCD: primary ciliary dyskinesia.)

**Table 1 diagnostics-13-03598-t001:** Patient data for each disease group.

	EOM	GPA	PCD	*p*-Value
Age, mean (SD), years	55.7 (7.7)	53.9 (17.9)	27.0 (17.6)	0.009 *^,a^
Male, *n* (%)	3 (33.3)	5 (41.7)	3 (50.0)	0.810
Bilateral otitis media, *n* (%)	8 (88.9)	4 (33.3)	5 (100.0)	0.004 *
Nasal polyp, *n* (%)	8 (88.9)	2 (16.7)	5 (83.3)	0.001 *
Chronic sinusitis, *n* (%)	9 (100.0)	11 (91.7)	6 (100.0)	0.545
Patients, *n*	9	12	6	

^a^: ANOVA test; *: *p* < 0.05; EOM: eosinophilic otitis media; GPA: granulomatosis with polyangiitis; PCD: primary ciliary dyskinesia.

**Table 2 diagnostics-13-03598-t002:** Patient data for the eosinophilic otitis media group and OME with ventilation tube insertion.

	EOM	OME
Age, mean (SD), years	55.7 (7.7)	55.4 (14.3)
Male, *n* (%)	3 (33.3)	122 (54.2)
Mean eosinophil concentration, % (SD)	12.10 (4.23)	2.63 (2.70)
>6, *n* (%)	9 (100)	23 (10.2)
≤6, *n* (%)	0 (0)	202 (89.8)
Mean eosinophil count (/μL)	850 (265.7)	162.8 (216.1)
>450, *n* (%)	9 (100)	13 (5.8)
≤450, *n* (%)	0 (0)	212 (94.2)
Total, *n* (%)	9 (100)	225 (100)

EOM: eosinophilic otitis media; OME: otitis media with effusion.

**Table 3 diagnostics-13-03598-t003:** Radiological characteristics of disease groups.

Involvement/Pneumatization	EOM	GPA	PCD	*p*-Value
Protympanum, %	88.2	55.6	75.0	0.096
Retrotympanum, %	47.1	50.0	100.0	0.029 *
Protympanum only, %	41.2	5.6	0	0.008 *
Eustachian tube, %	58.8	38.9	37.5	0.424
Mastoid cavity, %	35.3	61.1	100	0.009 *
Pneumatization, mean grade (SD)	3.1 (0.8) ^c^	2.4 (1.2) ^b^	1.1 (0.4) ^b,c^	<0.001 *^,a^
Ears, *n*	17	18	8	

A higher pneumatization grade (from 1 to 4) indicates a larger mastoid cavity; ^a^: Kruskal–Wallis test; ^b,c^: significant pairs in Dunn’s post hoc test (*p* = 0.013 and < 0.001 in ^b^ and ^c^, respectively); *: *p* < 0.05; EOM: eosinophilic otitis media; GPA: granulomatosis with polyangiitis; PCD: primary ciliary dyskinesia.

## Data Availability

The data presented in this study are available upon request from the corresponding author. The data are not publicly available due to ethical restrictions.

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
