# Peer review of "Diagnosis of Eosinophilic Otitis Media Using Blood Eosinophil Levels"

_diagnostics, 2023, doi:10.3390/diagnostics13233598_

Round 1

Reviewer 1 Report

Comments and Suggestions for Authors

Dear Authors,

This study was conducted using a retrospective medical chart review that tried to use blood eosinophil levels for the differential diagnosis of eosinophilic otitis media (EOM) from other conditions and to underline the clinical and radiologic characteristics are analyzed between EOM, granulomatosis with polyangiitis (GPA), and primary ciliary dyskinesia (PCD)

It is logically constructed and sets out the importance of an adequate differential diagnosis and the necessity of middle ear sampling before encountering it in the surgical field

The available research is well presented and discussed, and the conclusion is supported by the evidence presented

All the bibliographic data are very well inserted in this article, maybe quite a few, even the EOM is a rare disorder.

This study respects the methodology but had some limitations – the control group include only a few patients (27) compared to control group. It’s may be useful to do discuss also the differential diagnosis of eosinophilia.

More attention on the paper design (caps letter on the beginning of sentences)

Good luck

Author Response

This study was conducted using a retrospective medical chart review that tried to use blood eosinophil levels for the differential diagnosis of eosinophilic otitis media (EOM) from other conditions and to underline the clinical and radiologic characteristics are analyzed between EOM, granulomatosis with polyangiitis (GPA), and primary ciliary dyskinesia (PCD)

It is logically constructed and sets out the importance of an adequate differential diagnosis and the necessity of middle ear sampling before encountering it in the surgical field

The available research is well presented and discussed, and the conclusion is supported by the evidence presented

All the bibliographic data are very well inserted in this article, maybe quite a few, even the EOM is a rare disorder.

This study respects the methodology but had some limitations – the control group include only a few patients (27) compared to control group. It’s may be useful to do discuss also the differential diagnosis of eosinophilia.

We agree with your comment. We added a sentence to the limitation section as below.

Line 247.

“There are several limitations of this study. The disease groups had a small number of patients due to the low incidence of EOM, GPA, and PCD. Given the result of this study, blood eosinophils can be a crucial marker for differential diagnosis of EOM from other diseases since EOM has distinctive blood eosinophilia.

More attention on the paper design (caps letter on the beginning of sentences)

As you pointed out, we corrected several typos throughout our paper.

Good luck

------------------------------------------------------

Reviewer 2 Report

Comments and Suggestions for Authors

Really good research done. I agree as mentioned by authors, the need for a bigger sample size and prospective design to gain more information about EOM and to get a better understanding of the disease for optimization of the treatment.

Author Response

Thank you for reviewing our paper. We are currently collecting more patients for a future study.

Reviewer 3 Report

Comments and Suggestions for Authors

1.     There are too many circles in Figure 1, please use dot or simplify the circles. 

2.     Appendix A is not clear, please revise. 

3.     The citation format should comply with the rule of journal. [] should be put before period. 

4.     In conclusion, capitalize the beginning of the sentence.

5.     Please add the reference to the sentence “EOM has typical clinical characteristics that are significantly frequent compared to ordinary otitis media, which are associated with bronchial asthma in 90%, resistance to conventional treatment in 93%, viscous middle ear effusion in 93%, and nasal polyposis in 62% of EOM patients”.

6.     The simple size and small population in each group is too small to make the any possible conclusion. 

Comments on the Quality of English Language

Minor editing of English language required.

Author Response

  1. There are too many circles in Figure 1, please use dot or simplify the circles. 

We revised Figure 1 and the figure legend as below.

Figure 1. Peripheral blood eosinophil concentration and count of each disease group. Bars and error bars indicate median and range, respectively. Grey box indicates interquartile range.

  1. Appendix A is not clear, please revise. 

We revised Appendix A as you suggested.

  1. The citation format should comply with the rule of journal. [] should be put before period. 

As you suggested, we modified the format of all citations throughout our manuscript.

  1. In conclusion, capitalize the beginning of the sentence.

As you pointed out, we corrected the capitalization error.

  1. Please add the reference to the sentence “EOM has typical clinical characteristics that are significantly frequent compared to ordinary otitis media, which are associated with bronchial asthma in 90%, resistance to conventional treatment in 93%, viscous middle ear effusion in 93%, and nasal polyposis in 62% of EOM patients”.

As you suggested, we added a reference for the sentence you mentioned above.

‘EOM has typical clinical characteristics that are significantly frequent compared to ordi-nary otitis media, which are associated with bronchial asthma in 90%, resistance to con-ventional treatment in 93%, viscous middle ear effusion in 93%, and nasal polyposis in 62% of EOM patients [1].

  1. The simple size and small population in each group is too small to make the any possible conclusion. 

We agree with your comment, and we mentioned the small number of patients as a main limitation of our study. We also added a sentence highlighting the worth of this study despite its small size, as below.

Line 247.

“There are several limitations of this study. The disease groups had a small number of patients due to the low incidence of EOM, GPA, and PCD. Given the result of this study, blood eosinophils can be a crucial marker for differential diagnosis of EOM from other diseases since EOM has distinctive blood eosinophilia.

Minor editing of English language required.

We had our entire paper proofread and edited again by a professional English editor.